# DIFFERENTIALLY PRIVATE OPTIMIZATION FOR NON-DECOMPOSABLE OBJECTIVE FUNCTIONS

**Weiwei Kong, Andrés Muñoz Medina & Mónica Ribero**
Google Research
New York, NY, USA
{`weiweikong, ammedina, mribero`}@google.com

## ABSTRACT

Unsupervised pre-training is a common step in developing computer vision models and large language models. In this setting, the absence of labels requires the use of similarity-based loss functions, such as contrastive loss, that favor minimizing the distance between similar inputs and maximizing the distance between distinct inputs. As privacy concerns mount, training these models using differential privacy has become more important. However, due to how inputs are generated for these losses, one of their undesirable properties is that their $L_2$ sensitivity grows with the batch size. This property is particularly disadvantageous for differentially private training methods, such as DP-SGD. To overcome this issue, we develop a new DP-SGD variant for similarity based loss functions — in particular, the commonly-used contrastive loss — that manipulates gradients of the objective function in a novel way to obtain a sensitivity of the summed gradient that is $O(1)$ for batch size $n$. We test our DP-SGD variant on some CIFAR-10 pre-training and CIFAR-100 finetuning tasks and show that, in both tasks, our method's performance comes close to that of a non-private model and generally outperforms DP-SGD applied directly to the contrastive loss.

## 1 INTRODUCTION

Foundation models — large models trained in an unsupervised manner to be fine-tuned on specific tasks — have become one of the cornerstones of modern machine learning. These models generally outperform other approaches in multiple tasks, ranging from language generation, to image classification and speech recognition. In fact, models such as LaMDA (Thoppilan et al., 2022), BERT (Devlin et al., 2019), GPT (Radford et al., 2018) and diffusion models (Saharia et al., 2022; midjourney) interact with millions of users per day. Due to the complexity of these models, there are multiple concerns in the privacy community that these models may *memorize* some of the training data. For models trained on user-generated content, this may result in a catastrophic privacy breach, where the model may unintentionally reveal private information about a user. Recent work from Shokri et al. (2017) and Balle et al. (2022) showed that these risks are not just a theoretical concern and that it is possible to (i) know whether a particular example was in a dataset for training the model and (ii) reconstruct training data using only black-box access to the model.

Differential privacy provides an information-theoretic guarantee that the model does not depend drastically on any example (Dwork et al., 2006) and the aforementioned work also showed that these attacks become significantly harder when models are trained using differential privacy.

Consequently, private training methods have received considerable attention from the privacy community in the past decade. Some of the foundational work on this area was established by Chaudhuri et al. (2011) which provided algorithms for private learning with convex loss functions and Abadi et al. (2016) which proposed the differentially private stochastic gradient descent (DP-SGD) algorithm for privately training neural networks. Multiple lines of work have stemmed from this research area, ranging from tighter privacy analysis (Ghazi et al., 2022) to more efficient implementations of DP-SGD (Li et al., 2021). However, most of the literature on private machine learning makes one crucial assumption about the objective function they are trying to minimize: the objective decomposes as a sum (or average) of example level losses. This assumption drastically simplifies the

*sensitivity analysis* (how the objective changes as one changes one point in the dataset) of DP-SGD algorithm.

In this work, we focus on models that are trained using non-decomposable objective functions. That is, a function that cannot be described as a sum (or average) of individual losses. Our study is motivated by the use of contrastive losses (Oord et al., 2018; Chen et al., 2020a;b) for pre-training foundation models. Contrastive losses generally compare each example against all other examples in the batch and adding or removing an example to a batch of examples can affect the objective function in unpredictable ways. This type of behavior generally makes it hard, if not impossible, to train models privately. In this work, we show that common non-decomposable losses have a crucial property that makes them amenable to private training. Our contributions are summarized as follows:

- We provide a general framework for measuring the sensitivity of DP-SGD for certain non-decomposable losses.

- We show how to apply this framework to two common non-decomposable losses: contrastive loss and spreadout (regularization) loss (Zhang et al., 2017).

- We conduct experiments on privately pre-training large image classification models (a generic embedding model and Resnet18) and show that we can achieve performance comparable to non-private pre-training. Our experiments analyze the performance of simple pre-training as well as fine tuning on a downstream task.

## 2 PRELIMINARIES

*Notation.* Denote $[n] := \{1, \ldots, n\}$, $\mathbb{R}$ to be the set of real numbers, and $\mathbb{R}^d = \mathbb{R} \times \cdots \times \mathbb{R}$ where the Cartesian product is taken $d$ times.

Given a feature space $\mathcal{X}$, such as a space of images or sentences, we focus on unsupervised learning of embedding models $\Phi_w : \mathcal{X} \to \mathbb{R}^d$ parametrized by $w \in \mathcal{W}$ where $\mathcal{W}$ is a parameter space $\mathcal{W} \subset \mathbb{R}^p$.

Let $X = \{(x_i, x_i')\}_{i=1}^n \subseteq \mathcal{X} \times \mathcal{X}$ be a batch with $n$ records, such that $x_i$ and $x_i'$ are similar (positive pairs) in the feature space. These positive pairs can correspond, for instance, to two version of the same image, a sentence and its translation on a different language or an image and its caption. Let $S : \mathbb{R}^d \times \mathbb{R}^d \to \mathbb{R}$ be a function measuring similarity of two points in $\mathbb{R}^d$. A common objective is to find a parameter $w \in \mathcal{W}$ that preserves the similarities defined by pairs in $X$.

Given vectors $x_1, \ldots, x_n \in \mathbb{R}^d$, define $\mathbf{x} = (x_1, \ldots, x_n)$, and denote their embeddings as $\Phi_w(\mathbf{x}) = (\Phi_w(x_1), \ldots, \Phi_w(x_n))$.

Given embeddings $u, v_1, v_2, \ldots, v_n \in \mathbb{R}^d$, define the similarity profile of $u$ respect to $v_1, \ldots, v_m$ for $m \leq n$ as the vector $\mathbf{S}^m(u, \boldsymbol{v})$ of similarities between $u$ and the first $m$ vectors in $\boldsymbol{v}$. Formally, $\mathbf{S}^m(u, \boldsymbol{v}) \in \mathbb{R}^m$ where entry $j \in [m]$ is defined as $[\mathbf{S}^m(u, \boldsymbol{v})]_j = S(u, v_j)$. A common similarity function is the cosine similarity given by

$$[\mathbf{S}^m_{\cos}(u, \mathbf{v})]_j = \left\langle \frac{u}{\|u\|}, \frac{v_j}{\|v_j\|} \right\rangle \quad \forall j \in [n].$$

Given a dataset $X = \{(x_i, x_i')\}_{i=1}^n$ and a family of loss functions $\ell^{(i,n)} : \mathbb{R}^n \to \mathbb{R}$ that calculate the loss on the similarity profile of point $x_i$ based on the $n$ points on batch $X$, define

$$
\begin{aligned}
Z_X^{(i,n)}(w) &:= \mathbf{S}^n(\Phi_w(x_i), \Phi_w(x_1'), \ldots, \Phi_w(x_n')), \\
L_X^{(i,n)}(w) &:= \ell^{(i,n)} \circ Z_X^{(i,n)}(w), \\
\mathcal{L}_X(w) &:= \sum_{i=1}^n L_X^{(i,n)}(w),
\end{aligned}
\tag{1}
$$

for $i \in [n]$. The similarity terms $Z_X^{(1,n)}(w), \ldots, Z_X^{(n,n)}(w) \in \mathbb{R}^n$ are commonly referred to as contrastive logits. Given $\eta > 0$, contrastive loss models, which aim to minimize $\mathcal{L}_X(w)$, are typically

trained iteratively using stochastic gradient descent (SGD) as follows:

$$w_+ = w - \eta \nabla \mathcal{L}_X(w).$$

*Organization.* The rest of this section reviews some key concepts used to develop our proposed scheme. Section 3 reviews related works. Section 4 gives the main technical results and the proposed scheme that implements DP-SGD for general contrastive losses. For brevity, we leave the proof of these results for the Appendix at the end of this paper. Section 5 presents numerical experiments on CIFAR10 and CIFAR100, as well as a brief discussion on numerical bottlenecks. Finally, Section 6 gives a few concluding remarks.

## 2.1 DIFFERENTIAL PRIVACY

Let $\mathcal{Z}$ denote an arbitrary space and let $D = \{z_1, \ldots, z_n\} \subset \mathcal{Z}$ denote a dataset. We say that datasets $D$ and $D'$ are neighbors if $D' = D \cup \{z_{n+1}\}$ for some $z_{n+1} \in \mathcal{Z}$. A mechanism $M \colon \mathcal{Z}^* \to \mathcal{O}$ is a randomized function mapping a dataset to some arbitrary output space $\mathcal{O}$. Let $\epsilon, \delta > 0$. We say that mechanism $M$ is $(\epsilon, \delta)$-differentially private (Dwork et al., 2006) if for all neighboring datasets $D, D'$ and all $S \subset \mathcal{O}$ the following inequality holds:

$$P(M(D) \in S) \le e^\epsilon P(M(D') \in S) + \delta.$$

A simple way of ensuring that a mechanism is differentially private is by using the following process.

**Definition 2.1** (Gaussian Mechanism)**.** *Let $\epsilon, \delta > 0$ and $f \colon \mathcal{Z}^* \to \mathbb{R}^d$, and denote $\Delta_2(f) := \|f(D) - f(D')\|_2$ to be the $L_2$-sensitivity of the function $f$. For $\xi \sim \mathcal{N}(0, \sigma)$, the mechanism defined by*

$$M(D) = f(D) + \Delta_2(f)\xi,$$

*is $(\epsilon, \delta)$-differentially private for an appropriate[1] choice of $\sigma$.*

Our primary goal in this paper is to implement a Gaussian mechanism for the function $X \mapsto \nabla \mathcal{L}_X(w)$, where $\mathcal{L}_X(w)$ is as in (1).

## 2.2 LOSS FUNCTIONS

**Definition 2.2** (Canonical contrastive loss)**.** *The (canonical) contrastive loss function is given by $\mathcal{L}_X(w)$ in (1) with $\ell^{(i,n)}(Z) = -\log(e^{Z_i}/\sum_{j=1}^n e^{Z_j})$.*

The above loss essentially treats the unsupervised learning problem as a classification problem with $n$ classes, where the pair $(x_i, x_i')$ has a positive label and $(x_i, x_j')$ has a negative label for all $j \ne i$.

Contrastive loss is widely used by the vision community (Oord et al., 2018; Chen et al., 2020a;b; Radford et al., 2021) and has been shown to be extremely successful at obtaining pre-trained models for image classification.

**Definition 2.3** (Spreadout regularizer loss)**.** *The spreadout regularizer loss is given by $\mathcal{L}_X(w)$ in (1) with $\ell^{(i,n)}(Z) = \sum_{j \ne i} Z_j^2/(n-1)$.*

The spreadout regularizer is commonly used when training embedding models for computer vision (Zhang et al., 2017; Yu et al., 2020), used as a method to promote orthogonality in the embedding space among dissimilar objects in the whole feature space.

**Definition 2.4** (Summed loss from per-example loss)**.** *Let $Z = \{(x_1, y_1), \ldots, (x_n, y_n)\}$ be a dataset of features and label pairs. Given a set of per-example loss functions $\{f_Z^i\}_{i=1}^n$ corresponding to the examples in $Z$, the summed loss function is $\mathcal{K}_Z(w) = \sum_{i=1}^n f_Z^i(w)$.*

## 2.3 NAIVE CLIPPING SCHEMES

Before presenting our scheme, we discuss some naive approaches for bounding the sensitivity of contrastive loss gradients during DP-SGD training.

---

[1]See, for example, Balle & Wang (2018).

We first review how DP-SGD is typically applied for the summed loss $\nabla \mathcal{K}_Z(w)$ in (2.4). It can be shown that the precise $L_2$-sensitivity of $\nabla \mathcal{K}_Z(w)$ in DP-SGD is generally hard to estimate in deep learning settings (Latorre et al., 2020; Shi et al., 2022). As a consequence, for given a $L_2$-sensitivity bound $B$ on $\nabla \mathcal{K}_Z(w)$, practitioners usually clip the per-example gradients $\nabla f_Z^i(w)$ by the bound $B$ and apply the Gaussian mechanism on the sum of the clipped gradients to obtain the differentially private (DP) gradient that is passed to DP-SGD. This is motivated by the fact that adding or removing an example from the dataset $Z$ will not change the norm of the DP gradient (and, hence, its sensitivity) by more than $B$. Also, notice that the standard deviation of the Gaussian mechanism's noise is $B\sigma$ which is independent of the sample size $n$.

Let us now compare the above results with the $L_2$-sensitivity of a similar scheme for contrastive loss functions $\mathcal{L}_X(w)$ as in (1). For neighboring datasets $X = \{(x_i, x_i')\}_{i=1}^n$ and $X^\circ = \{(x_i, x_i')\}_{i=1}^{n-1}$, the sensitivity of $\mathcal{L}_X(w)$ is given by

$$\|\nabla \mathcal{L}_X(w) - \nabla \mathcal{L}_{X^\circ}(w)\| = \left\| \nabla L_X^{(n,n)}(w) + \sum_{i=1}^n \left[ \nabla L_X^{(i,n)}(w) - \nabla L_{X^\circ}^{(i,n-1)}(w) \right] \right\|, \quad (2)$$

where $L_X^{(i,n)}(w)$ is as in (1). Similar to the per-example loss, for a given $L_2$-sensitivity bound $B$, we *could* consider clipping the "per-example" gradient terms $\{\nabla L_X^{(i,n)}(w)\}_{i=1}^n$ (for DP-SGD on dataset $X$) and $\{\nabla L_{X^\circ}^{(i,n-1)}(w)\}_{i=1}^{n-1}$ (for DP-SGD on dataset $X^\circ$) by $B$ and applying the appropriate Gaussian mechanism. However, applying the triangle inequality to the bound in (2), the $L_2$-sensitivity of the resulting scheme is $O(nB)$. As a consequence, the standard deviation of the Gaussian mechanism's noise is $O(nB\sigma)$ which is $O(n)$ worse than for per-example losses.

As another alternative (Huai et al., 2020; Kang et al., 2021), one could directly clip $\nabla \mathcal{L}_X(w)$ or $\nabla \mathcal{L}_{X^\circ}(w)$ by $B$ and apply the Gaussian mechanism to these clipped gradients with a standard deviation of $O(B\sigma)$ (see (2)). However, we show in our experiments section that this approach, nicknamed *Naive-DP*, does not materially reduce the value of $\mathcal{L}_X(w)$, even when varying the batch size or the clip norm value.

> *Our proposed scheme aims to provide the first DP-SGD scheme which materially reduces the loss value $\mathcal{L}_X(w)$ without requiring a dependence on the batch size $n$ in the underlying Gaussian mechanism's noise.*

## 3    RELATED WORK

Contrastive learning has had large impact on unsupervised pretraining of computer vision models (Chen et al., 2020a;b) and representation learning for language models (Logeswaran & Lee, 2018; Chidambaram et al., 2018), or both (Radford et al., 2021). Fang et al. (2020); Giorgi et al. (2020); Wu et al. (2020) use a contrastive loss function for pre-training and fine-tuning BERT with data augmentation. More recently it has been used for reinforcement learning with BERT-type models (Banino et al., 2021).

In the private setting, the majority of the work has been focused on improving the original implementation of DP-SGD (Abadi et al., 2016) for decomposable losses. Research has particularly focused on tighter privacy guarantees on DP-SGD via advanced privacy accounting methods (Mironov, 2017; Ghazi et al., 2022) or solving computational issues, for example associated with gradient clipping (Goodfellow, 2015), or improving a specific models efficiency and scalability such as privately pre-training T5 (Ponomareva et al., 2022). For non-decomposable losses, some researchers have studied private learning from pairwise losses in the convex and strongly convex case (Huai et al., 2020; Xue et al., 2021) and test only in a diabetes dataset. Later works (Yang et al., 2021; Kang et al., 2021) obtain similar results for the non-convex case; these approaches circumvent clipping by assuming access to the Lipschitz constant of the loss function, which depends on the encoder function (typically a deep neural network). However, this Lipschitz constant is generally not easy to estimate (Latorre et al., 2020; Shi et al., 2022).

Xu et al. (2022) learn private image embeddings with user-level differential privacy, but avoid unsupervised training and, consequently, avoid non-decomposable loss functions such as contrastive and triplet loss. Instead, this work relays a supervised multi-class classification problem, and avoids

dependencies among different records, at the cost of labeling the data. Similarly, Yu et al. (2023) train ViP, a foundation model for computer vision but replace the contrastive (non-decomposable) loss with an instance-separable loss. Li et al. (2022) propose noising the similarity matrix between pairs of inputs and compute a noisy loss function. They combine this with a noisy gradient but assume a per-gradient bounded sensitivity.

Orthogonal works study attacks to embedding models. For instance Song & Raghunathan (2020) showed that when trained without differential privacy, embedding models can be inverted. More specifically, model attacks are able to recover members from the training set when the attack are designed to recover information from embeddings trained with a contrastive loss (Liu et al., 2021). To prevent specific attacks Rezaeifar et al. (2022) developed an architecture that learns an obfuscator that prevents reconstruction or attribute inference attacks. He & Zhang (2021) quantifies the exposure risk under contrastive learning losses and develops an adversarial training procedure to mitigate the risk. However, none of these approaches provide differential privacy guarantees. Finally, Wu et al. (2022) explores contrastive learning in federated settings, where users feed a user-embedding; the negative samples are created at the server with the differentially private embeddings sent by the users.

## 4  BOUNDING PAIRWISE-CONTRIBUTIONS

This section first introduces a condition on the family of loss functions $\{\ell^{(i,n)}\}$ that, when combined with a clipping operation on the gradient of the similarity between each pair of records, permits the derivation of a DP-SGD variant that benefits from increasing the batch size when using similarity based loss functions.

We start by deriving an expression for the gradient of $\mathcal{L}$ in Lemma 4.1 that highlights the dependence on the gradient of pairwise similarity values $S(\Phi_w(x_i), \Phi_w(x'_j))$. By leveraging this decomposition, we find a bound on the overall loss $\mathcal{L}$ gradient's sensitivity in Theorem 4.2. Finally, we combine these two facts to produce a differentially private optimization algorithm for similarity based loss functions. We defer proofs to the supplementary material.

### 4.1  COMPUTING GRADIENT SENSITIVITY

Lemma 4.1 below shows that the gradient of a similarity based loss function can by expressed in terms of the pairwise similarity gradients $\nabla_w S(\Phi_w(x_i), \Phi_w(x'_j))$.

**Lemma 4.1.** *Let $\mathcal{L}_X(w)$ and $Z_X^{(i,n)}(w)$ be as in* (1), *and denote*

$$Z_X^i(w) := Z_X^{(i,n)}(w),$$
$$Z_X^{ij}(w) := [Z_X^{(i,n)}(w)]_j = S(\Phi_w(x_i), \Phi_w(x'_j)).$$

*Then,*

$$\nabla_w \mathcal{L}_X(w) = \sum_{i=1}^n \sum_{j=1}^n \frac{\partial \ell^{(i,n)}(Z_i(w))}{\partial Z_X^{ij}} \nabla Z_X^{ij}(w). \tag{3}$$

We now describe conditions on the family $\{\ell^{(i,n)}\}$ function that allow us to derive a bound on the $L_2$-sensitivity of $\nabla \mathcal{L}_X(w)$.

**Theorem 4.2.** *Let $\mathcal{L}_X(w)$ and $Z_X^{ij}$ be as in Lemma 4.1, let $\mathcal{C} \subseteq \mathbb{R}$ be a compact set, and let $\mathbf{z}' \in \mathcal{C}^{n-1}, z_n \in \mathcal{C}$, and $\mathbf{z} = (\mathbf{z}', z_n) \in \mathcal{C}^n$. Assume that for all $i \in [n]$ the family of functions $\{\ell^{(i,n)}\}_{(i,n) \in \mathbb{N} \times \mathbb{N}}$ satisfies*

$$\sum_{j=1}^{n-1} \left| \frac{\partial \ell^{(i,n)}(\mathbf{z})}{\partial z_j} - \frac{\partial \ell^{(i,n-1)}(\mathbf{z}')}{\partial z_j} \right| \leq L, \quad \sum_{j=1}^n \left| \frac{\partial \ell^{(i,n)}(\mathbf{z})}{\partial z_j} \right| \leq G_1, \quad \sum_{i=1}^n \left| \frac{\partial \ell^{(i,n)}(\mathbf{z})}{\partial z_n} \right| \leq G_2, \tag{4}$$

*where $L, G_1$, and $G_2$ can depend on $n$. If $\|Z_X^{ij}\|_2 \leq B$ for every $i$ and $j$, then the $L_2$-sensitivity of $\nabla \mathcal{L}_X(w)$ can be bounded as*

$$\Delta_2(\nabla \mathcal{L}_X) \leq (G_1 + G_2 + (n-1)L)B.$$

We are now ready to present our main algorithm. Before proceeding, the following two corollaries show that one can obtain a private estimate of the gradient of the training loss by clipping the pairwise similarity gradients and applying a Gaussian mechanism.

**Corollary 4.3.** *Let $B > 0$, $z \in \mathbb{R}^d$, $\mathcal{L}_X(w)$ and $Z_X^{ij}$ be as in Lemma 4.1, and let $\mathrm{Clip}_B(x) := \min\{B/\|x\|, 1\} x$ denote the vector $x$ clipped to have norm at most $B$. If the family of functions $\{\ell^{(i,n)}\}_{i=1}^n$ satisfy the conditions of Theorem 4.2. Then, the function*

$$\mathbf{X} \mapsto \sum_{i=1}^n \sum_{j=1}^n \frac{\partial \ell^{(i,n)}(Z_i(w))}{\partial Z_X^{ij}} \mathrm{Clip}_B(\nabla Z_X^{ij}(w)) \tag{5}$$

*has $L_2$ sensitivity bounded by $(G_1 + G_2 + (n-1)L)B$.*

**Corollary 4.4.** *If the family of loss functions $\ell^{(i,n)}$ satisfies the conditions of Theorem 4.2, then each iteration of Algorithm 1 satisfies $(\epsilon, \delta)$-differential privacy[2] for $\epsilon = \sqrt{\log(1.25/\delta)}/\sigma$.*

*Proof.* The proof is immediate since each step of the algorithm corresponds to the Gaussian mechanism with noise calibrated to the sensitivity of the mechanism. □

Moreover, in the following lemmas, we present how condition (4) holds with $L = O(1/n)$ for the contrastive and spreadout regularizer losses under a cosine similarity. Consequently, this ensures that the $L_2$-sensitivity given by (2) is independent of $n$.

**Lemma 4.5.** *(Contrastive loss) Let $\ell^{(i,n)}$ be as in Definition 2.2 with $\mathbf{S}^n = \mathbf{S}_{\cos}^n$. Then $\ell^{(i,n)}$ satisfies the conditions of Theorem 4.2 with*

$$G_1 + G_2 + (n-1)L \leq 2 \left( 1 + \frac{(n-2)e^2}{e^2 + (n-1)} \right). \tag{6}$$

**Lemma 4.6.** *(Spreadout loss) Let $\ell^{(i,n)}$ be as in Definition 2.3 with $\mathbf{S}^n = \mathbf{S}_{\cos}^n$. Then $\ell^{(i,n)}$ satisfies the conditions of Theorem 4.2 with $G_1 + G_2 + (n-1)L \leq 6$.*

## 4.2 MAIN ALGORITHM

We present Logit-DP, our proposed DP-SGD scheme in Algorithm 1. The algorithm specifically receives a batch size $n$, learning rate (or schedule) $\eta$, a number of iterations $T$, constants $G_1$, $G_2$, and $L$ defined in Theorem 4.2, and the similarity gradient clip norm $B$. It then computes the sensitivity of the overall gradient $C$ (line 2).

The algorithm proceeds to the training loop where, at each iteration $t$, it samples a batch of size $n$. Then, instead of per-example gradients, it computes similarity gradients $g_{ij}$ (line 6), clips all $g_{ij}$ vectors to obtain a bounded gradients $\bar{g}_{ij}$, and computes an approximate gradient for $\mathcal{L}$ using (5) (line 9). Finally, it applies noise (line 9) and updates the model (line 10).

While algorithm 1 uses SGD as the gradient step, the model update in line 10 can be passed to other gradient based optimizers such as Adagrad (McMahan & Streeter, 2010; Duchi et al., 2011) or Adam (Kingma, 2014). Remark that all previous work on privacy accounting for DP-SGD also applies to our algorithm as each iteration simply generates a private version of the gradient of the batch loss.

## 5 NUMERICAL EXPERIMENTS

This section presents numerical experiments that compare the practical viability of our proposed DP-SGD variant (Logit-DP), the implementation of DP-SGD (Naive-DP) which clips the aggregated gradient at the batch level, and non-private SGD (Non-Private). Specifically, we examine several training and testing metrics on pre-training and fine-tuning tasks applied to the CIFAR10 and CIFAR100 datasets using a generic embedding net model and a ResNet18 model without batch

---

[2] A slightly tighter relation between $\sigma$ and $\epsilon$ can be given using the results on the analytic Gaussian mechanism of Balle & Wang (2018).

---

**Algorithm 1:** Logit-DP

---

**Input:** Sensitivity bound $B > 0$, sensitivity constants $G_1, G_2, L > 0$, dataset
$D = \{(x_i, x'_i)\}_{i=1}^N$, batch size $n$, iteration limit $T \geq 1$, stepsize $\eta > 0$, noise multiplier
$\sigma > 0$, model $\Phi$

**Output:** Embedding model $\Phi_{w_T}$

1  Initialize weights $w_0$ in $\Phi$;
2  Compute gradient sensitivity $C = (G_1 + G_2 + nL)B$;
3  **for** $t = 1, 2, ..., T - 1$ **do**
4    Sample batch $X = \{(x_1, x'_1), ..., (x_n, x'_n)\}$;
5    **for** $i, j = 1, ..., n$ **do**
6      Compute similarity gradients $\nabla Z_X^{ij}(w_t) = \nabla_{w_t} S(\Phi_{w_t}(x_i), \Phi_{w_t}(x'_j))$;
7      Clip gradients to obtain $\text{Clip}_B(\nabla Z_X^{ij}(w_t)) = \min\left\{\frac{B}{\|\nabla Z_X^{ij}(w_t)\|}, 1\right\} \nabla Z_X^{ij}(w_t)$;
8    **end**
9    Compute $\bar{g}$ using (5)  Compute noisy gradient $\tilde{g} = \bar{g} + Y$ with $Y \sim \mathcal{N}(0, \sigma C I_p)$;
10   Update the model $w_{t+1} = w_t - \eta\tilde{g}$;
11 **end**

---

normalization layers, as their standard implementation isn't privacy-preserving; in table 3 in the appendix we show the effect of removing these layers. All DP methods chose a noise multiplier so that $\epsilon$-DP is achieved for $\epsilon = 5.0$. The details of the embedding models, the hyperparameters of the each variants, and the training setups for each task are given in the supplementary material. Our code is publicly available at https://github.com/google-research/google-research/tree/master/logit_dp

The last subsection describes strategies to manage memory requirements encountered as training scales to larger models and datasets.

## 5.1 Pre-training on CIFAR10

In these experiments, all DP-SGD and SGD variants were given model $\Phi$, which was either a generic embedding model or a ResNet18 model without batch normalization layers. Each variant was tasked with minimizing the contrastive loss described in Example 2.2 for the examples in the CIFAR10 dataset. For testing/evaluation metrics, we examined the quality of the embedding model under a $k$-nearest neighbors ($k$-NN) classifier for $k = 3$.

Figure 1 presents the observed (relative) training loss values over the number of examples seen so far for ten different training runs using the generic embedding model and the effect of batch size on Naive-DP. In particular, the plot in Figure 1 demonstrates that Naive-DP's loss value is mostly unchanged for large batch sizes and noisy for small batch sizes. Table 1 presents the relative averaged test metrics at the last evaluation point.

Similar trends to Figure 1 were observed for the ResNet18 model.

Table 1: Relative aggregate CIFAR10 test metrics generated by the confusion matrix $C$ at the last test point over ten runs. Each aggregate metric is divided by the corresponding one for Non-Private. Aggregate accuracy is defined as $\sum_i C_{ii} / \sum_{i,j} C_{ij}$ averaged over all runs. The recall, precision, and $F_\beta$ scores are the average of the best observed metric over all ten CIFAR10 classes.

|  | Embedding Net Metrics | | | | ResNet18 Metrics | | | |
|---|---|---|---|---|---|---|---|---|
|  | Accuracy | Recall | Precision | $F_\beta$ Score | Accuracy | Recall | Precision | $F_\beta$ Score |
| Logit-DP | 0.819 | 0.855 | 0.812 | 0.831 | 0.730 | 0.871 | 0.695 | 0.768 |
| Naive-DP | 0.827 | 0.827 | 0.812 | 0.820 | 0.599 | 0.672 | 0.699 | 0.685 |
| Non-private | 1.000 | 1.000 | 1.000 | 1.000 | 1.000 | 1.000 | 1.000 | 1.000 |

For additional reference, we have the following figures and tables in the supplementary material. Each variant's confusion matrices at the last evaluation point are in Figures 3–4. The (absolute) means and standard deviations of the test metrics at the last evaluation point are in Table 3. Finally,

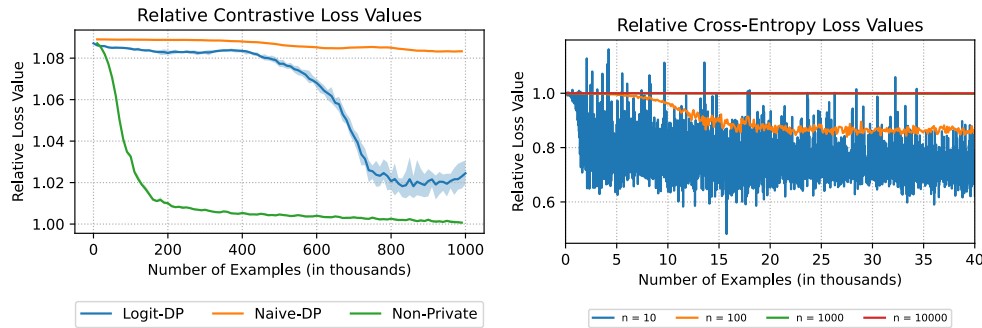

Figure 1: (Left) Relative CIFAR10 training loss over ten runs. Relative loss is defined as the observed training loss divided by the minimum loss observed across all runs and all variants. Shaded regions bound the observed loss values over the runs, while the the dark lines represent the average relative loss observed so far. (Right) Single runs of Naive-DP with the same settings as in the left graph but with different batch sizes $n$. The $n = 1000$ and $n = 10000$ form mostly overalapping lines.

the relative training loss over runtime and the training speed over number of examples seen is given in Figure 5.

## 5.2 FINE-TUNING ON CIFAR100

Pre-trained foundational models are often non-privately fine-tuned on classification tasks for local use and, consequently, are not required to be privately optimized. In these experiments, we test the ability of the privately pre-trained embedding model to adapt to new tasks. All variants were given the generic embedding model $\Phi$ from Subsection 5.1 and a multilayer fully-connected model $\Psi$. They were then tasked with non-privately minimizing the cross-entropy loss generated by the combined model $\Phi \circ \Psi$ on the CIFAR100 dataset to predict the coarse label of the input (20 categories), under the condition that the weights in $\Phi$ were frozen, i.e., could not be updated.

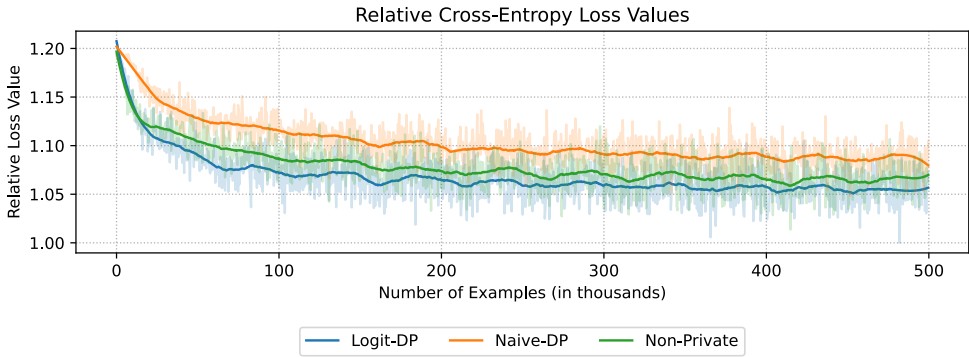

Figure 2: Relative CIFAR100 training loss for a single run. Relative loss is defined as the observed training loss divided by the minimum loss observed across all variants. Lightly colored lines are the true loss values, while the dark lines are smoothed loss values generated by a third-order Savitzky-Golay filter with a sliding window of 100 observations.

For reference, we present each variant's (absolute) test metrics — at the last evaluation point — in Table 4 of the supplementary material.

## 5.3 A MEMORY BOTTLENECK AND A POTENTIAL FIX

In our implementation of Logit-DP (Algorithm 1), a computational bottleneck was the materialization of the $n^2$ logit gradients ($g_{ij}$ in Algorithm 1) for a batch $n$ examples, which were needed to compute the final aggregated gradient ($\bar{g}$ in Algorithm 1). A potential solution is to compute gra-

Table 2: Relative CIFAR100 test metrics generated by the confusion matrix $C$ at the last test point over one run. Each metric is divided by the corresponding one for non-private SGD. Accuracy is defined as $\sum_i C_{ii} / \sum_{i,j} C_{ij}$ while top recall, precision, and $F_\beta$ scores are the best observed metric over all CIFAR100 classes.

| | Embedding Net Metrics | | | |
| --- | --- | --- | --- | --- |
| | Accuracy | Recall | Precision | $F_\beta$ Score |
| Logit-DP | 1.013 | 0.969 | 0.954 | 0.981 |
| Naive-DP | 0.946 | 1.296 | 0.665 | 0.911 |
| Non-private | 1.000 | 1.000 | 1.000 | 1.000 |

dients $g_{ij}$ sequentially. While addressing the memory bottleneck, this solution is computationally inefficient in terms of runtime.

Below, we describe an alternative approach for computing $\bar{g}$ and argue that it is more efficient for certain choices $\Phi_w$. Consider the function

$$F_X(w) := \sum_{i=1}^n \sum_{j=1}^n \lambda_{ij} \mathbf{S}^n(\Phi_w(x_i), \Phi_w(x'_j))$$

where $\lambda_{ij}$ are fixed, real-valued weights given by

$$\tau_{ij} := \frac{\partial \ell^{(i,n)}}{\partial Z_{ij}}(\mathbf{S}^n(\Phi_w(x_i), \Phi_w(x'_j))), \quad \lambda_{ij} := \tau_{ij} \min\left\{\frac{B}{\|g_{ij}\|}, 1\right\} \quad \forall i, j,$$

and note that $\bar{g} = \nabla F_X(w)$ (cf. (5)). In view of the previous identity, an alternative approach to computing $\bar{g}$ is to first compute each $\lambda_{ij}$ and *then* compute $\nabla F_X(w)$.

This new approach has the following advantages: (i) given $\lambda_{ij}$, the memory and runtime cost of computing the gradient of $F_X(w)$ is on the same order of magnitude as computing the gradient of $\mathcal{L}(w, X) = \sum_{i=1}^n \ell^{(i,n)}(\mathbf{S}^n(\Phi_w(x_i), \Phi_w(\mathbf{x}')))$ when both methods employ backpropagation, (ii) the memory cost of storing the weights $\lambda_{ij}$ is only $\Theta(n^2)$, and (iii) the costs of computing the weights $\lambda_{ij}$ requires only computing the $n^2$ scalar pairs $(\tau_{ij}, \|g_{ij}\|)$ rather than computing the $n^2$ gradients $g_{ij}$ of size $|w|$ as in Algorithm 1.

The last advantage is of particular interest, as there are well-known methods Goodfellow (2015); Lee & Kifer (2020); Rochette et al. (2019) in the literature to efficiently computing the norms $\|g_{ij}\|$ without materializing each $g_{ij}$. For example, some of these methods decompose $g_{ij}$ into a low-rank representation $g_{ij} = U_{ij} V_{ij}^\intercal$ for low-rank matrices $U_{ij}$ and $V_{ij}$, and then exploit the identity

$$\|g_{ij}\|^2 = \|U_{ij} V_{ij}^\intercal\|^2 = \langle U_{ij}^\intercal U_{ij}, V_{ij}^\intercal V_{ij} \rangle.$$

When $U_{ij}$ and $V_{ij}$ are column vectors, the last expression above reduces to $\|U_{ij}\|^2 \|V_{ij}\|^2$, which can be substantially more efficient than first materializing $g_{ij} = U_{ij} V_{ij}^\intercal$ and then computing $\|g_{ij}\|$. A correct implementation of this technique is far from trivial, and we leave this as future research.

## 6 CONCLUDING REMARKS

As observed in Section 3, naive implementations of DP-SGD for similarity-based losses are ineffective because the standard deviation of the noise in the Gaussian mechanism grows with $n$. Experiments in the previous section show that even with careful hyperparameter tuning, the loss remains nearly constant during pre-training. These results are even more pronounced for Resnet18, when the number of model parameters is large. Fine-tuned models using Naive-DP also perform less effectively compared to both the non-private baseline and the Logit-DP algorithm.

Careful analysis of these losses and their decomposition shows that by clipping logit gradients, Logit-DP obtains a sensitivity that is constant on the batch size, considerably reducing the magnitude of the noise added to privatize the gradient. These insights expand the suite of tasks that can be trained in a privacy preserving way with only marginal drops in accuracy. Work on more efficient implementations of these algorithms is an interesting avenue of future work and we introduced several concrete ideas at the end of the previous section.

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

SUPPLEMENTARY MATERIAL FOR DP-SGD FOR NON-DECOMPOSABLE
OBJECTIVE FUNCTIONS

## A  PROOFS

*Proof of Lemma 4.1.* Denote

$$Z_i^\circ := [Z_{i1}^\circ, \dots, Z_{i(n-1)}^\circ] := \mathbf{S}^{n-1}(\Phi_w(x_i), \Phi_w(\mathbf{x}')) \in \mathbb{R}^{n-1}.$$

for all $i$. Using the chain rule, the gradient of $\mathcal{L}_X(w)$ on batch $\{(x_i, x_i')\}_{i=1}^n$ can be computed as

$$\nabla \mathcal{L}_X(w) = \sum_{i=1}^n \nabla \ell^{(i,n)}(Z_X^i(w))^\top DZ_X^i(w). \tag{7}$$

The conclusion now follows by combining the above expression with the fact that $DZ_X^i(w) = [(\nabla Z_X^{i1})^\top; \cdots; (\nabla Z_X^{in})^\top]$. $\qquad\square$

*Proof of Theorem 4.2.* For ease of notation, let $Z_X^i = Z_X^i(w)$, $Z_X^{ij} = Z_X^{ij}(w)$, $\ell_X^{(i,n)} = \ell^{(i,n)}(Z_X^i)$, and

$$\ell_X^{(i,n-1)} = \ell^{(i,n-1)}(Z_X^{i1}, \dots, Z_X^{i(n-1)}),$$

and similarly for the gradients of these functions in $w$. Using Lemma 4.1, the $\ell_2$ sensitivity of $\nabla \mathcal{L}_X$ is

$$\Delta_2(\nabla \mathcal{L}_X) = \left\| \sum_{i=1}^n \sum_{j=1}^n \frac{\partial \ell_X^{(i,n)}}{\partial Z_X^{ij}} \nabla Z_X^{ij} - \sum_{i=1}^{n-1} \sum_{j=1}^{n-1} \frac{\partial \ell_X^{(i,n-1)}}{\partial Z_{ij}} \nabla Z_X^{ij} \right\|.$$

The above expression can be broken down into the following terms:

$$\Delta_2(\nabla \mathcal{L}_X) = \left\| \underbrace{\sum_{j=1}^n \frac{\partial \ell_X^{(n,n)}}{\partial Z_X^{nj}} \nabla Z_X^{nj}}_{T1} + \underbrace{\sum_{i=1}^{n-1} \frac{\partial \ell_X^{(i,n)}}{\partial Z_X^{in}} \nabla Z_X^{in}}_{T2} + \underbrace{\sum_{i=1}^{n-1} \sum_{j=1}^{n-1} \left( \frac{\partial \ell_X^{(i,n)}}{\partial Z_X^{ij}} - \frac{\partial \ell_X^{(i,n-1)}}{\partial Z_X^{ij}} \right) \nabla Z_X^{ij}}_{T3} \right\|.$$

We now use the triangle inequality to bound each term.

$$\|T_1\| = \left\| \sum_{j=1}^n \frac{\partial \ell_X^{(n,n)}}{\partial Z_X^{nj}} \nabla Z_X^{nj} \right\| \le \sum_{j=1}^n \left| \frac{\partial \ell_X^{(n,n)}}{\partial Z_X^{nj}} \right| \left\| Z_X^{nj} \right\| \sum_{j=1}^n \left| \frac{\partial \ell_X^{(n,n)}}{\partial Z_X^{nj}} \right| B \le G_1 B$$

Similarly, using the same approach, we obtain $\|T_2\| \le G_2 B$. Finally, using the assumption on the partial derivatives of the family $\{\ell^{(i,n)}\}_{n \in \mathbb{N}}$

$$\begin{aligned}
\|T_3\| &= \left\| \sum_{i=1}^{n-1} \sum_{j=1}^{n-1} \left( \frac{\partial \ell_X^{(i,n)}}{\partial Z_X^{ij}} - \frac{\partial \ell_X^{(i,n-1)}}{\partial Z_X^{ij}} \right) \nabla Z_X^{ij} \right\| \\
&\le \sum_{i=1}^{n-1} \sum_{j=1}^{n-1} \left| \frac{\partial \ell_X^{(i,n)}}{\partial Z_X^{ij}} - \frac{\partial \ell_X^{(i,n-1)}}{\partial Z_X^{ij}} \right| \left\| \nabla Z_X^{ij} \right\| \\
&\le \sum_{i=1}^{n-1} \sum_{j=1}^{n-1} \left| \frac{\partial \ell_X^{(i,n)}}{\partial Z_X^{ij}} - \frac{\partial \ell_X^{(i,n-1)}}{\partial Z_X^{ij}} \right| B \\
&\le (n-1) B L
\end{aligned}$$

Combining the above bounds yields the desired bound on $\Delta_2(\nabla \mathcal{L}_X)$.

$\qquad\square$

*Proof of Lemma 4.5.* It is straightforward to show that

$$\frac{\partial \ell^{(i,n)}(z)}{\partial z_j} = \begin{cases} e^{z_i}/\sum_{k=1}^{n} e^{z_k} - 1, & \text{if } j = i, \\ e^{z_j}/\sum_{k=1}^{n} e^{z_k}, & \text{otherwise.} \end{cases}$$

It then follows that

$$\sum_{j=1}^{n} \left| \frac{\partial \ell^{(n,n)}(z)}{\partial z_j} \right| = \sum_{j=1}^{n-1} \frac{e^{z_j}}{\sum_{k=1}^{n} e^{z_k}} + 1 - \frac{e^{z_n}}{\sum_{k=1}^{n} e^{z_k}}$$

Note that $\sum_{j=1}^{n-1} e^{z_j}/\sum_{k=1}^{n} e^{z_k} = 1 - e^{z_n}/\sum_{k=1}^{n} e^{z_k}$, since the $n$ terms constitute a probability distribution summing up to 1. Thus,

$$\sum_{j=1}^{n} \left| \frac{\partial \ell^{(n,n)}(z)}{\partial z_j} \right| = 1 - \frac{e^{z_n}}{\sum_{k=1}^{n} e^{z_k}} + 1 - \frac{e^{z_n}}{\sum_{k=1}^{n} e^{z_k}} = 2\left(1 - \frac{e^{z_n}}{\sum_{k=1}^{n} e^{z_k}}\right)$$

Denoting $p_n = e^{z_n}/\sum_{k=1}^{n} e^{z_k}$, we have that

$$G_1 = \sum_{j=1}^{n} \left| \frac{\partial \ell^{(n,n)}(z)}{\partial z_j} \right| = 2\left(1 - p_n\right). \tag{8}$$

Next, we look into $G_2 = \sum_{i=1}^{n-1} |\partial \ell^{(i,n)}(z)/\partial z_n|$. In this case,

$$\sum_{i=1}^{n-1} \left| \frac{\partial \ell^{(i,n)}(z)}{\partial z_n} \right| = \sum_{i=1}^{n-1} \frac{e^{z_n}}{\sum_{k=1}^{n} e^{z_k}} = \frac{(n-1)e^{z_n}}{\sum_{k=1}^{n} e^{z_k}}$$

and, hence,

$$G_2 = (n-1)p_n. \tag{9}$$

Finally, for the condition on the difference we have that for all $i \in [n]$,

$$L = \sum_{j=1}^{n-1} \left| \frac{\partial \ell^{(i,n-1)}(z)}{\partial z_j} - \frac{\partial \ell^{(i,n)}(z)}{\partial z_j} \right|$$

$$= \sum_{j=1}^{n-1} \left| \frac{e^{z_j}}{\sum_{k=1}^{n-1} z_k} - \frac{e^{z_j}}{\sum_{k=1}^{n} z_k} \right| = \sum_{j=1}^{n-1} \frac{e^{z_j}}{\sum_{k=1}^{n-1} z_k} - \frac{e^{z_j}}{\sum_{k=1}^{n} z_k},$$

where we removed the absolute value on the last term since all values are positive. We observe that the first term sums up to 1, and the last one corresponds to $1 - e^{x_n}/\sum_{k=1}^{n} e^{x_k} = 1 - p_n$. Hence, the above expression is given by

$$L = 1 - \sum_{j=1}^{n-1} \frac{e^{z_j}}{\sum_{k=1}^{n} e^{z_k}} = 1 - (1 - p_n) = p_n. \tag{10}$$

Combining (8), (9), and (10), we have

$$G_1 + G_2 + (n-1)L = 2(1 - p_n) + (n-1)p_n + (n-1)p_n = 2(1 + (n-2)p_n).$$

Since the contrastive loss uses the cosine similarity, we have that the inputs $(z_1, \ldots, z_n)$ given to $\ell^{(i,n)}$, satisfy $|z_i| \le 1$ for all $i$. Consequently, the last term is maximized when $z_n = 1$ and $z_k = -1$ for $k \le n-1$, yielding

$$G_1 + G_2 + (n-1)L \le 2\left(1 + \frac{(n-2)e}{e + (n-1)e^{-1}}\right) = 2\left(1 + \frac{(n-2)e^2}{e^2 + (n-1)}\right),$$

where we multiplied by $e$ the numerator and denominator on the last step. The result follows from theorem 4.2. □

*Proof of Lemma 4.6.* Let us calculate the values of $L, G_1, G_2$ that make this function satisfy the conditions of Theorem 4.2. Let $\boldsymbol{z} = (\boldsymbol{z}', z_n)$ then we have that

$$\frac{\partial \ell^{(i,n)}(\boldsymbol{z})}{\partial z_j} - \frac{\partial \ell^{(i,n-1)}(\boldsymbol{z}')}{\partial z_j} = \frac{2}{n-1}z_j - \frac{2}{n-2}z_j = -\frac{2}{(n-1)(n-2)}z_j,$$

for $j \neq i$ and 0 otherwise. For any $C \leq \max_{j=1\ldots n} z_j$, we then have:

$$\sum_{j=1}^{n-1}\left|\frac{\partial \ell^{(i,n)}(\boldsymbol{z})}{\partial z_j} - \frac{\partial \ell^{(i,n-1)}(\boldsymbol{z}')}{\partial z_j}\right| \leq \sum_{j \neq i}^{n-1}\frac{2}{(n-1)(n-2)}|z_j| \leq \frac{2C}{n-1} =: L$$

Similarly we have

$$\sum_{j=1}^{n}\frac{\partial \ell^{(n,n)}(\boldsymbol{z})}{\partial z_j} \leq 2C =: G_1, \quad \sum_{i=1}^{n}\frac{\partial \ell^{(i,n)}(\boldsymbol{z})}{\partial z_j} \leq 2C \qquad =: G_2$$

and, hence, $G_1 + G_2 + (n-1)L \leq 6C$. Since the cosine similarity implies $z_{ij} = \mathrm{S}(\Phi(x_i), \Phi(x'_j)) \leq 1$, it holds that $C = 1$ and the result follows. $\qquad\square$

# B EXPERIMENT DETAILS

This appendix gives more details about the numerical experiments in Section 5. All models were trained on a single NVidia V100 GPU using a cloud computing platform with 512 GB of RAM.

## B.1 PRE-TRAINING ON CIFAR10

**Model Specification, Dataset Details, and Hyperparameters**

For reproducibility, we now give the details of the model, the hyperparameters of the above variants, and the training setup. The generic embedding net model consists of three 2D convolution layers followed by one embedding layer. The convolution layers used a 3-by-3 kernel with a stride of 2, a ReLU output activation function, a Kaiming-normal kernel initializer, and (sequentially) chose output channels of 8, 16, and 32, respectively. The embedding layer generated an output of dimension 8 and used a Xavier-normal initializer.

The learning rates for Logit-DP, Naive-DP, and Non-Private were $10^{-2}$, $10^{-2}$, and $10^{-3}$, respectively, for the generic embedding net experiments and $10^{-4}$, $10^{-3}$, and $10^{-2}$, respectively, for the ResNet18 experiments. All variants used the standard Adam optimizer for training and used the canonical 80-20 train-test split of the CIFAR10 dataset. However, Logit-DP used 25 and 100 gradient accumulation steps for the generic embedding net and ResNet18 experiments, respectively. The batch size during training was $10,000$ and $1,000$ for the generic embedding net and ResNet18 experiments, respectively, and the entire testing dataset was used for evaluating test metrics. Moreover, each variant was run for 20 and 2 epochs over the entire training dataset for the generic embedding net and ResNet18 experiments in Table 1, respectively.

For the DP variants, we fixed the desired $\ell_2$ sensitivity to be $10^{-4}$ and $10^{-5}$ for Naive-DP and Logit-DP, respectively, in the generic embedding net experiments and $10^{-3}$ and $10^{-5}$, respectively, in the ResNet18 experiments. All DP methods chose a noise multiplier so that $\varepsilon$-DP was achieved for $\epsilon = 5.0$.

Finally, all hyperparameter tuning was done through a grid search of various learning rates ($10^{-5}$, $10^{-4}$, ..., $10^{-2}$) and $\ell_2$ sensitivities ($10^{-6}$, $10^{-5}$, ..., $10^{-0}$).

## B.2 FINE-TUNING ON CIFAR100

**Model Specification, Dataset Details, and Hyperparameters**

For reproducibility, we now give the details of the model, the hyperparameters of the above variants, and the training setup. $\Psi$ is a three-layer fully-connected neural network whose layer output dimensions are 64, 32, and 20 in sequence.

The learning rate for all variants was $10^{-2}$. All variants used the standard Adam optimizer (iteration scheme) for training and used the canonical 80-20 train-test split of the CIFAR100 dataset. The batch size during training was $400$ and the entire testing dataset was used for evaluating test metrics. Moreover, each variant was run for ten epochs over the entire training dataset.

For the DP variants, we fixed the desired $\ell_2$ sensitivity to be 1.0 and chose a noise multiplier so that $\varepsilon$-DP was acheived for $\epsilon = 5.0$. All hyperparameter tuning was done through a grid search of various learning rates ($10^{-4}$, $10^{-3}$, $10^{-2}$) and $\ell_2$ sensitivities ($10^{-2}$, $10^{-1}$, $10^{0}$).

## C    ADDITIONAL FIGURES AND TABLES

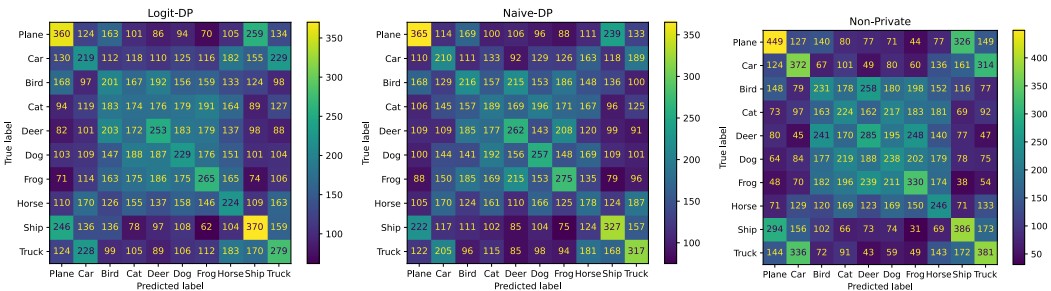

Figure 3: Averaged CIFAR10 confusion matrices at the last testing step for the generic embedding net experiments. Values are rounded down to the nearest whole number.

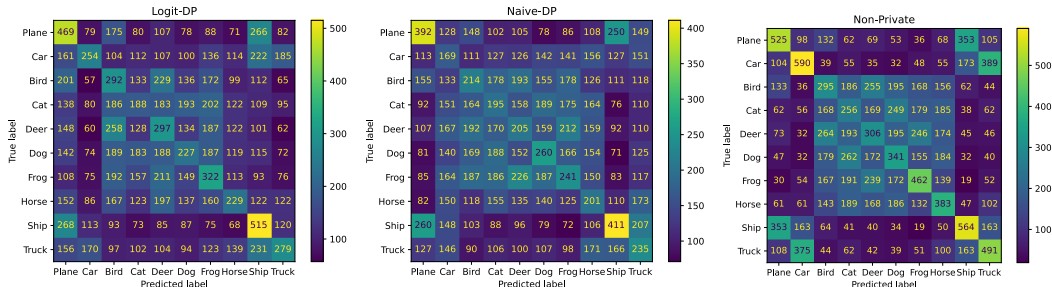

Figure 4: Averaged CIFAR10 confusion matrices at the last testing step for the ResNet18 experiments. Values are rounded down to the nearest whole number.

| | Embedding Net Metrics (mean / standard deviation) | | | |
| --- | --- | --- | --- | --- |
| | Accuracy | Recall | Precision | $F_\beta$ Score |
| Logit-DP | 0.173 / 0.002 | 0.254 / 0.006 | 0.251 / 0.005 | 0.253 / 0.004 |
| Naive-DP | 0.174 / 0.002 | 0.242 / 0.005 | 0.245 / 0.006 | 0.243 / 0.005 |
| Non-private | 0.212 / 0.003 | 0.300 / 0.010 | 0.312 / 0.010 | 0.306 / 0.010 |
| | ResNet18 Metrics (mean / standard deviation) | | | |
| | Accuracy | Recall | Precision | $F_\beta$ Score |
| Logit-DP | 0.202 / 0.006 | 0.325 / 0.013 | 0.268 / 0.013 | 0.291 / 0.013 |
| Naive-DP | 0.169 / 0.003 | 0.269 / 0.006 | 0.284 / 0.010 | 0.276 / 0.008 |
| Non-private (- BN) | 0.278 / 0.008 | 0.389 / 0.013 | 0.388 / 0.011 | 0.388 / 0.010 |
| Non-private (+ BN) | 0.274 / 0.004 | 0.402 / 0.016 | 0.418 / 0.015 | 0.41 / 0.014 |

Table 3: Aggregate CIFAR10 test metrics generated by the confusion matrix $C$ at the last test point over ten runs. Accuracy is defined as $\sum_i C_{ii} / \sum_{i,j} C_{ij}$. The recall, precision, and $F_\beta$ scores are over the best observed metric over all ten CIFAR10 classes. Non-private (+BN) and Non-Private (-BN) denote the standard and modified architecture without BatchNorm layers respectively.

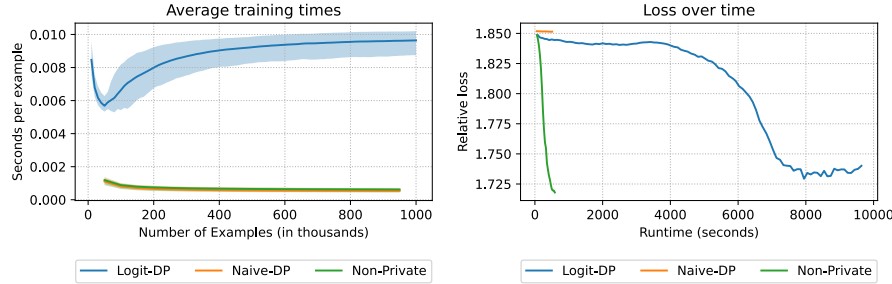

Figure 5: Training time related plots for the small embeddding net model on CIFAR10 over ten runs. (Left) Number of seconds per example over the number of examples seen. Shaded regions bound the observed values, while the dark lines represent the averaged values. (Right) Average training losses over the average runtime.

| | Embedding Net Metrics | | | |
| --- | --- | --- | --- | --- |
| | Accuracy | Recall | Precision | $F_\beta$ Score |
| Logit-DP | 0.169 | 0.432 | 0.308 | 0.336 |
| Naive-DP | 0.158 | 0.578 | 0.215 | 0.313 |
| Non-private | 0.167 | 0.446 | 0.322 | 0.343 |

Table 4: CIFAR100 test metrics generated by the confusion matrix $C$ at the last test point over one run. Accuracy is defined as $\sum_i C_{ii} / \sum_{i,j} C_{ij}$ while top recall, precision, and $F_\beta$ scores are the best observed metric over all CIFAR100 classes.

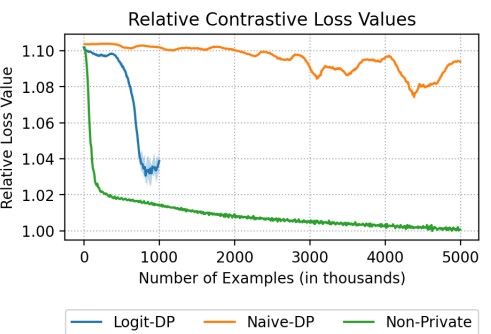

Figure 6: Relative loss on CIFAR10 over 100 epochs. This extended training run (cf. Figure 1, left) demonstrates that the performance of Logit-DP is not solely due to early stopping.

