# OpenReview forum: "Differentially private optimization for non-decomposable objective functions"
_ICLR.cc/2025/Conference — ICLR 2025 Poster_

### Official Review · Reviewer_BSWK · 2024-10-28

**Soundness:** 3
**Presentation:** 3
**Contribution:** 2
**Rating:** 5
**Confidence:** 3

**Summary:**

This paper presents a variant of DPSGD tailored for DP-training with non-decomposable loss functions, such as the CLIP loss. The authors theoretically bound the l2 sensitivity of the gradient, demonstrate the privacy guarantee, and introduce their Logit-DP algorithm. They also discuss prior methods like per-batch and per-sample clipping and evaluate their approach on CIFAR-10 pretraining and CIFAR-100 finetuning tasks.

**Strengths:**

The problem addressed is both important and practically valuable, and the method provides an interesting solution for DP-training models with non-decomposable losses, such as CLIP-based VLMs.

**Weaknesses:**

See below.

**Questions:**

1. The main result, Theorem 4.2, seems to be a simple expansion with some triangle inequlities. Much of the theoretical complexity is deferred to parameters L, G1, and G2, which seem to be technical artifacts rather than providing meaningful generalization of the theory.

2. The algorithm involves per-sample clipping, computing the batch gradient using Theorem 4.2, and adding noise. This approach somehow feels somewhat redundant to me, as it seems to enforce clipping of the per-sample loss explicitly. Since per-batch clipping alone should suffice to ensure privacy (see [1]), it seems that per-sample clipping is not an essential requirement from a privacy perspective, but rather a technical necessity due to the proof requiring bounded per-sample gradients. While the authors discuss the empirical limitations of per-sample clipping (see 5 below), it would be helpful if the authors could further clarify the reasoning behind this choice, particularly any theoretical intuitions beyond this empirical justification.

3. The DP guarantee in Corollary 4.4 also seems loose. The noise magnitude of order $n \sqrt{\log(1/\delta)} / \epsilon$ could be reduced to $\sqrt{\log(1/\delta)} / \epsilon$, similar to [1], given that the loss is not divided by $1/n$, while the loss in [1] is divided by $1/n$.

4. The paper lacks discussion on the privacy-utility tradeoff. While noise can always be added to ensure privacy, it’s crucial to evaluate its impact on model performance.

5. In Sections 5.1 and 5.2, the values of $\epsilon$ is missing (maybe I missed it...?) Additionally, the authors suggest that “clipping the batch gradient (which they term Naive-DP) does not significantly reduce $L_X(w)$, regardless of batch size or clip norm.” They imply that Naive-DP leads to poor convergence: excessive noise for small batches and stability issues for large batches. However, is there a rationale beyond Figure 1? Also, was the learning rate kept constant across different batch sizes? Typically, larger batch sizes requires larger learning rates (see [2]).

6. Lastly, in Figure 2, Logit-DP performs better than the non-private model, which seems counterintuitive.


[1] Huang et al. Safeguarding Data in Multimodal AI: A Differentially Private Approach to CLIP Training, https://arxiv.org/pdf/2306.08173
[2] Ponomareva et al. How to DP-fy ML: A Practical Guide to Machine Learning with Differential Privacy, https://arxiv.org/pdf/2303.00654

---

> ### Author Response · Authors · 2024-11-20
> **Response to reviewer BSWK (part 1)**
>
> Thanks for the review!
>
> 1. While our proof indeed utilizes triangle inequalities, our core contribution lies in the careful decomposition of the terms. This decomposition, achieved through a combination of the chain rule and well-chosen inequalities, enables to bound specific terms balancing the sensitivity while maintaining more signal to noise ratio than NaiveDP. This improvement is not merely a technical artifact; it has real implications for practical applications.
>
>     Moreover, we demonstrate this improvement both theoretically and empirically. Corollaries 4.5 and 4.6 provide theoretical guarantees for the performance gains, while our experiments further validate these findings in practice. Furthermore, Theorem 4.2 is not just a specific technical result. It generalizes to a broader class of non-decomposable loss functions (unexplored by the DP community), extending the classic DP-SGD with per-example clipping. This generalization is a significant contribution, as it expands the applicability of differential privacy to a wider range of machine learning models and tasks.
>
> 2.  The reviewer may have missed Example 2.3., where we detail why batch clipping is not efficient with contrastive losses. Indeed, per-example (or per-pairwise) clipping is not the only way of achieving privacy since the sensitivity can be more easily bounded with batch clipping. However, per-example clipping is essential to obtain high utility private models. As we note in Section 2.3., summed losses noise scales as n with batch clipping and only constantly with our procedure.  Empirically this is also observed in our experiments, where naive-DP performs batch clipping.
>
> 3.  Thanks for raising this subtlety. We work with summed losses as this notation is more easy and is conventional in the DP community (See Bassily et al 2014).
>
>     More crucially, this convention highlights the advantages of our method. When the objective function is expressed as the average of per-example losses, batch clipping adds constant noise while the noise magnitude of our approach decreases with larger batch sizes as n grows. This scaling is a key benefit of our technique, leading to improved performance with large batches.
>
>     It's also worth noting that while the sensitivity (and thus the noise magnitude) can change depending on the formulation of the loss function, the differential privacy guarantee itself remains unaffected.
>
> 4. Thanks for raising this concern. This tradeoff is a central theme throughout our paper. Our core contribution is precisely addressing this tradeoff by proposing a clipping method that significantly reduces the amount of noise required for a given privacy guarantee.
> We demonstrate this improvement both theoretically and empirically:
>
>    - Theoretically: Our sensitivity analysis shows that our clipping scheme has constant sensitivity, while traditional batch clipping increases linearly with the batch size n. This directly translates to a significant reduction in the standard deviation noise for the same privacy guarantee.
>
>    - Empirically: Our experiments show that our method achieves significantly better utility compared to naive DP (batch clipping) for the same privacy level. In some cases, batch clipping leads to catastrophic performance degradation, highlighting the severity of the privacy-utility tradeoff with existing methods.

---

> > ### Author Response · Authors · 2024-11-20
> > **Response to reviewer BSWK (part 2)**
> >
> > 5. We have added $\epsilon$ values to the revised manuscript (line). We used $\epsilon=5$ for all private methods. We did perform a hyperparameter sweep across learning rates, batch sizes, and L2 norm clip values, reporting the best result for each method. This ensures a fair comparison and accounts for the interaction between learning rate and batch size.
> >
> >     Regarding the performance of naive-DP (batch clipping), our observations about its poor convergence in Figure 1 are based on several factors:
> >
> >     -  Sensitivity Analysis: Our theoretical analysis demonstrates that the sensitivity of Naive-DP scales linearly with the batch size. This leads to a higher noise magnitude, hindering convergence, especially for large batches.
> >
> >     - Clip Value Tradeoff: Small clip values, while reducing noise, can severely restrict the gradient updates, slowing down convergence. Conversely, large clip values necessitate higher noise to maintain privacy, adding excessive variance to the training process. Crucially, increasing the batch size in Naive-DP does not improve this tradeoff. The noise magnitude remains high, limiting the potential benefits of larger batches.
> >
> >      In contrast, our method allows for a noise scale that decreases with the batch size. This leads to improved model performance as we can leverage larger batch sizes without incurring excessive noise.
> >
> > 6. In terms of cross-entropy loss, logit-DP does indeed perform better than non-private. However, Table 2 demonstrates that non-private still outperforms logit-DP in recall, precision and $F_\beta$ score. That is, logit-DP is not better than non-private across all dimensions.

---

### Official Review · Reviewer_chx4 · 2024-10-31

**Soundness:** 2
**Presentation:** 2
**Contribution:** 2
**Rating:** 3
**Confidence:** 4

**Summary:**

This papers discusses the challenge of training computer vision and language models with differential privacy, especially when using unsupervised pre-training and similarity-based loss functions like contrastive loss. The primary issue with contrastive loss in a privacy-preserving setting is that its sensitivity increases with the batch size, which negatively impacts differentially private methods like DP-SGD (Differentially Private Stochastic Gradient Descent). To address this, the authors propose a modified DP-SGD method specifically designed for similarity-based losses, achieving a constant sensitivity level regardless of batch size. Their experiments on CIFAR-10 and CIFAR-100 datasets demonstrate that this approach nearly matches the performance of non-private models and surpasses the results of directly applying DP-SGD to contrastive loss.

**Strengths:**

The paper considers developing DP algorithms for problems where the objective function is coupled, which often happens in problems with contrastive loss. The setting appears to be new, and the algorithms are specifically design and analyzed for this class.

**Weaknesses:**

However, the derivation of the \ell_2 sensitivity is rather trivial, and the resulting algorithm is still just a simple variation of the traditional DP-SGD algorithm. Most of the derivations are mechanical. I do not see too much insights or contribution in this work. Further, the paper has not been well-written, as there are places that have obvious typo and inconsistencies. For example in line 156 Eq. (2.4) has been mentioned, but no (2.4) has been defined, nor the gradient of K_z(w) has been defined either.

**Questions:**

n/a

---

> ### Author Response · Authors · 2024-11-20
> **Response to reviewer chx4**
>
> We appreciate the reviewer's feedback, although we respectfully disagree with their assessment of our work.
>
> The reviewer states that our derivations are trivial and that the resulting algorithm is a “simple” variation of DP-SGD.  However, the challenge of adapting differential privacy to non-decomposable loss functions (such as contrastive loss) has been a significant blocker in the field.  Previous research has often circumvented this issue altogether (Xu et al. 2022, Li et al. 2022, Yu et al 2023), potentially missing out on the advantages of unsupervised learning approaches.
>
> Our work directly addresses this challenge by carefully analyzing the derivative of the general non-decomposable losses and deriving its sensitivity. This analysis involves a non-trivial decomposition using the chain rule and the identification of specific conditions for generalizable non-decomposable loss functions. We then provide concrete examples and applications for widely used losses like the contrastive loss and the spreadout regularizer.
>
> Regarding our algorithm being a “simple” variation – while many DP breakthrough algorithms build upon existing mechanisms the specific application and subtleties of implementation can be complex.  For example, the exponential mechanism (See Dwork & Roth 2014) is an easy to describe mechanism, but sampling from it can be computationally hard. Similarly, the well known K-norm mechanism’s applicability is instance specific, and recent works (e.g. Joseph & Yu 2023) demonstrate the ongoing efforts in refining it for simple problems like sum and count queries. Our work makes a significant contribution by enabling the use of contrastive loss for training with sensitive data, opening up new possibilities in areas like medical image diagnostics.
>
> We encourage the reviewer to re-examine our work; we believe that a deeper analysis will reveal the non-trivial nature of our contributions.
>
> Additionally, we have carefully reviewed our manuscript and corrected the typographical errors and inconsistencies pointed out by the reviewer.
>
> Let us know if you have precise questions you would want us to address.

---

### Official Review · Reviewer_e6qK · 2024-11-04

**Soundness:** 2
**Presentation:** 2
**Contribution:** 2
**Rating:** 5
**Confidence:** 4

**Summary:**

This work studies DP optimization of non-decomposable functions in neural networks, trying to add less noise with respect to the batch size. A new algorithm with proofs are proposed with CIFAR experiments.

**Strengths:**

The paper is reasonably easy to follow and rigorous as far as I am concerned. A significant portion of the paper is devoted to background introduction. The empirical results are clear. The algorithmic limitation (the computation bottleneck) is carefully discussed.

**Weaknesses:**

The algorithm is not straightforward: $\bar g$ comes from (5) which does not relate to $\bar g$ explicitly nor to $\bar g_{ij}$ in the line above. There should be a hidden for loop between line 4 and 5 in Algorithm 1 (for i,j in 1...n) to highlight the computational bottleneck.

The experiments are not very convincing, especially given the first sentence in abstract "Unsupervised pre-training is a common step in developing computer vision mod- els and large language models." Some experiments are fine-tuning not pre-training, and no language modeling is shown. I would encourage the authors to enhance the experiments with ImageNet (at least a subset) or Transformers.

Minors:
1. Most new contents are only presented after page 4, which may come too late and is slightly distracting.

2. Title should not use abbreviations like DP and, since the method applies to other optimizers besides SGD, maybe the title can highlight this.

3. Line 3 of Algorithm 1, the for loop counting is incorrect.

**Questions:**

See "Weaknesses".

---

> ### Author Response · Authors · 2024-11-20
> **Response to reviewer e6qK**
>
> Thanks for the review!
>
> 1. Thanks for the suggestions, we’ve included the notation change into Algorithm 1 in the revised version.
>
> 2. We would like to clarify that our experiments indeed evaluate unsupervised private pre-training, both in terms of the pre-training objective itself (Section 5.1) and its impact on downstream tasks through fine-tuning (Section 5.2). Fine-tuning is a standard practice in evaluating pre-trained models, as it demonstrates the effectiveness of the learned representations on different downstream tasks.
>
> 3. We acknowledge the popularity of language models. While our work primarily focuses on computer vision, the theoretical contributions and the proposed method are applicable to a wide range of input data, including language modeling. We chose to focus on computer vision due to its significant impact in various fields that deal with sensitive data, such as medicine. We believe that demonstrating the effectiveness of our method in image data showcases its importance and broader applicability. We emphasize that our theoretical results hold for any type of input data, including those used in language models.
>
> **Minor.**
> 1. While we could provide a more comprehensive summary of our contributions, the content leading up to section 4 (notation, naive clipping, and example losses) is necessary to motivate the contributions.
>
> 2.  This is a great point, thanks for the recommendation, We’ve updated the title and added a comment in section 4.2.
> Line 3 of Algorithm 1, the for loop counting is incorrect.
>
> 3. Thank you for catching this. The corrected sequence is $t=1,2,\ldots,T-1$.

---

> > ### Comment · Reviewer_e6qK · 2024-11-27
> >
> > Thank you for taking time to respond. I agree the new revision reads better and clearer. I have decided to keep my score.

---

### Official Review · Reviewer_2dkT · 2024-11-04

**Soundness:** 2
**Presentation:** 2
**Contribution:** 2
**Rating:** 1
**Confidence:** 4

**Summary:**

This paper proposes a variant of DP-SGD for loss functions that cannot be described as an average (or sum) of individual losses, such as contrastive loss and spread-out regularization. The authors apply a Gaussian Mechanism scheme, theoretically derive an upper bound for $L_2$-sensitivity and experimentally claim their findings on computer vision tasks.

**Strengths:**

**S1:** The method is consistent with prior studies on DP-SGD.

**S2:** The implementation of the proposed Logit-DP algorithm is the same as for DP-SGD.

**S3:** The authors also prove several lemmas regarding the conditions on $L_2$-sensitivity.

**Weaknesses:**

**W1:** The main theoretical findings of the paper focus only on "vanilla" contrastive loss and spread-out loss. However, there is an increasing demand for more sophisticated types of loss functions, such as SimCLR [1] and InfoNCE [2]. These functions align well with **Definition 2.2** and benefit from a large batch size. The paper lacks additional commentary on these and other contrastive objectives, for which analogous lemmas can be easily proved.

**W2:** The experimental validation is unclear:

- In Line 326 the authors note that they train model without BatchNorm layers because of they are not privacy-preserving, at the same time, no comparison is provided with a baseline model that includes BatchNorm.
- Modern NLP and CV models often use transformer architectures that incorporate LayerNorm. Could the authors clarify whether they consider LayerNorm more privacy-preserving or not?
-  It is hard to understand figures: in Figure 1 the authors report averaged relative loss along with its bounds/percentiles for Logit-DP, but no bounds/percentiles provided for Non-Private and Naive-DP. Also there are no mentioned standard deviation in all Tables, while it is obvious that the authors report an averaged results.
- There are several drawbacks regarding the hyperparameters sweep: the authors train a ResNet18 model on CIFAR10 for only 2 epochs and a generic convolutional model for 20 epochs, whereas training from scratch on CIFAR10 typically requires hundreds of epochs. At the same time, they fine-tuned both the generic convolutional model and ResNet18 for 10 epochs, which is much longer than regular fine-tuning.
- In Table 4, the authors report an accuracy score below 20% for the CIFAR100 fine-tuning task. Needs further clarifications.

**W3:** In Lines 380-383, the authors do not address the issue of private fine-tuning of the publicly available (i.e., non-privately trained) model,  which is almost more popular in the community than private training from scratch. Additionally, the authors do not clarify what is meant by a ''privately pre-trained'' embedding model: how is it pre-trained?

**W4:** The code is not publicly available.


**Minor Comments:**

**C1:** There is no need to write ''non-private'' twice in Line 405,  as the implication is clear that the Non-Private optimizer is SGD (Line 324).

**C2:** The section title in Line 824 is incorrect; it should be  ''Fine-tuning on CIFAR100''.

**C3:** In Line 834 you denote $L_2$-sensitivity as $\ell_2$-sensitivity, which is not self-consistent.





[1] Ting Chen, Simon Kornblith, Mohammad Norouzi, Geoffrey Hinton. ''A Simple Framework for Contrastive Learning of Visual Representations''. ICML 2020

[2] Aaron van den Oord, Yazhe Li, Oriol Vinyals. ''Representation Learning with Contrastive Predictive Coding''.

**Questions:**

See Weaknesses.

---

> ### Author Response · Authors · 2024-11-20
> **Response to reviewer 2dkT (part 1)**
>
> Thanks for the review!
>
>  **W1.**
>
> Both SimCLR and InfoNCE are indeed within the scope of our analysis.
>   - SimCLR uses the canonical contrastive loss, which corresponds with Definition 2.2 in our paper. We already explicitly state this connection after the definition and provide sensitivity bounds in Lemma 4.5.
>   - InfoNCE: While incorporating a context variable, InfoNCE's core loss function and batch sampling strategy still fall under the framework of Definition 2.2.
>
>   We have included references to both SimCLR and InfoNCE in the Introduction, Related Work, and Preliminaries sections.
>
> **W2.**
>
>   - We understand the importance of thorough comparisons. However, we deliberately excluded BatchNorm layers from our models due to their inherent incompatibility with privacy preservation. Our focus is to rigorously analyze the impact of clipping and noise on unsupervised private learning. Introducing BatchNorm would blur the effects of these crucial components, making it difficult to isolate their individual contributions.
>
>  &ensp; &ensp; This focused approach provides insights into the core challenges of private contrastive learning. While a comparison with models that include BatchNorm layers might be interesting in other contexts, it falls outside the scope of our current paper.
>
>   - LayerNorm is not designed for privacy preservation. It normalizes across features within a single sample, ensuring that each sample is processed independently. This makes it easier to integrate with differential privacy techniques, as it avoids the aggregation of sensitive information across different samples. In contrast, BatchNorm normalizes a single feature across multiple samples in a batch. This aggregation of information across samples can potentially leak private information, making it more challenging to combine with differential privacy mechanisms.
>
>   &ensp;&ensp;&ensp; Privacy implications of both LayerNorm and BatchNorm are still being actively explored within the differential privacy  community (see [1, 2] below) , LayerNorm's sample-wise operation is inherently more compatible with DP techniques.
>
>  &ensp;&ensp;&ensp; [1] Davody, A., Adelani, D. I., Kleinbauer, T., & Klakow, D. (2020). On the effect of normalization layers on differentially private training of deep neural networks. arXiv preprint arXiv:2006.10919.
>
>  &nbsp;&nbsp; &nbsp; [2] Ponomareva, Natalia, et al. "How to dp-fy ml: A practical guide to machine learning with differential privacy." Journal of Artificial Intelligence Research 77 (2023): 1113-1201.
>
>   -  Bounds are actually provided for naive-DP and non-private but these bounds are miniscule when compared to logit-DP (e.g., zooming into the plots, you can see some shaded regions for non-private around the 180K examples mark).
>
> &nbsp;&nbsp; &nbsp; &nbsp;We have updated the absolute metrics in Appendix B with standard deviations.
>
>   -  Our primary objective in these experiments is to provide a clear demonstration of the theoretical and performance advantages offered by our proposed method. More specifically, we aim to showcase its potential for performance improvement within a constrained experimental setup. Consequently, the chosen stopping criteria for each experiment aimed to clearly demonstrate the loss improvement offered by our method compared to previous baselines across different settings. These results, even with a limited number of training epochs, effectively illustrate the performance boost achieved by our method.
>
> &ensp; &ensp; &ensp;  We acknowledge that our chosen training regime deviates from typical practices for training for several epochs. This  choice was deliberately made to prioritize efficiency in computational resources, especially given that this work focuses  on the validation of our theoretical contributions.
>
> &ensp; &ensp; &ensp; Moreover, note that the difference between the number of training epochs between pre-training and fine-tuning is not a direct comparison since pre-training utilizes different batch sizes optimized for each considered method. Larger batch sizes often allow for faster convergence, potentially requiring fewer epochs.
>
>   - We acknowledge that the accuracy scores for the CIFAR100 fine-tuning task are lower than typically observed. As mentioned above, our goal was to demonstrate the advantages of our proposed method, namely that we introduce a method that is able to overcome the loss stagnation of current contrastive loss methods, rather than training until state of the art results. This choice was to balance computational efficiency and save resources effectively, as it is clear from the plots that other methods are underperforming.

---

> > ### Author Response · Authors · 2024-11-20
> > **Response to reviewer 2dkT (part 2)**
> >
> > **W3.**
> >
> > Privately fine-tuning a publicly available model is a popular technique in the community. However, our focus in this work was on private training from scratch, which has been dismissed by the community because of the difficulties of DP optimization with contrastive losses and “solved” assuming the existence of public data being available.
> > In addition, it's important to clarify that fine-tuning typically involves labeled datasets and per-example loss functions like categorical cross-entropy, which are already well-studied in the context of differential privacy. Our work primarily focuses on addressing the challenges of privately training models with contrastive loss, which, again,  is more prevalent in self-supervised pre-training scenarios.
> >
> >   Regarding the "privately pre-trained" embedding model in Section 5.2, it refers to the model trained using our private contrastive learning method described in Section 5.1. We apologize for any lack of clarity on this point.
> >
> >
> > **W4.** We were planning on open sourcing the code but did not include a link for anonymity reasons. We will include in the next few days an anonymized version together with the supplemental material.
> >
> > **Minor comments.**  We addressed C1-C3 in the revised version.

---

> > > ### Comment · Reviewer_2dkT · 2024-11-21
> > >
> > > > **Minor comments.** We addressed C1-C3 in the revised version.
> > >
> > > C2 is still unaddressed.
> > >
> > >
> > > **Summary:**
> > >
> > > Overall, thank to the authors for the detailed response. They have clarified for me **W1**, and partially addressed **W2** (regarding the bound on the plot at 180K examples mark) and have added a part of my correction to the text.
> > >
> > > However, my concerns **1) - 5)**, regarding the meaningfulness of the experimental validation remain. The absence of at least one comparison with BatchNorm layers, one full training for 100+ epochs on CIFAR10, and one proper fine-tuning experiment make it difficult to fully trust the results of the experimental section.
> > >
> > > Without this kind of comparison, I will not be able to increase the current score.
> > > But I am looking forward to your new comments.

---

> > > > ### Author Response · Authors · 2024-11-22
> > > >
> > > > Thanks for the thoughtful feedback! We address your concerns point-by-point below, with particular attention to your questions regarding BatchNorm and the perceived limitations of our experimental setup.
> > > >
> > > > >1) BatchNorm itself is a layer that harms privacy-preserving training. From our experiments it is unclear whether you achieve a private training / fin-tuning because of the Logit-DP or the absence of batch normalization. To my opinion, at least one simple experiment on this matter should be adjusted.
> > > >
> > > > To ensure clarity, we have included in Table 3 results for both a standard ResNet with BatchNorm and the same model without BatchNorm. This highlights the impact of clipping, independent of BatchNorm removal.
> > > >
> > > > We would like to clarify that BatchNorm layers do not "harm" privacy-preserving training, but they are fundamentally incompatible with the privacy guarantees of differential privacy (DP).  BatchNorm relies on batch-specific statistics, which introduce privacy leakage.  Removing BatchNorm is necessary to maintain privacy during training.
> > > >
> > > > While DP-compatible BatchNorm layers are an interesting research direction [1,2], designing such layers requires sophisticated techniques for private computation of batch statistics and careful privacy accounting. This is outside the scope of our current work, which focuses on the novel Logit-DP clipping technique for non-decomposable objective functions which have received nearly no attention in the past.
> > > >
> > > > > “you achieve a private training / fin-tuning (...) because the absence of batch normalization.”
> > > >
> > > > As noted above, removing BatchNorm is necessary but not sufficient for private training. Logit-DP achieves privacy by carefully analyzing sensitivity and adding Gaussian noise during training, building upon established DP-SGD techniques. The improved utility compared to Naive-DP stems from our method's ability to reduce noise while maintaining privacy.
> > > >
> > > > > While reading your work, I noted that there is quite a wide class of models you could consider, at least for the image classification tasks, such as ViTs and alike architectures. It seems more natural to focus on these transformer-based models rather than using "truncated" versions of ResNets (without BatchNorm), especially if you have decided to exclude batch normalization altogether.
> > > >
> > > > Thanks for the recommendation, indeed ViT are high-performing vision models [3]. Note that this same paper [3] demonstrates that ResNets without BatchNorm achieve comparable results, with less than a 1% accuracy decrease even when compared to the best ViTs (see Section 4, experiments, table 5). This supports the validity of our choice of model architecture.
> > > >
> > > > Our primary goal is to demonstrate the effectiveness of Logit-DP and its improvement over current DP methods. We achieve this by showcasing its performance on a class of high-utility models.  Exploring the application of Logit-DP on state-of-the-art architectures like ViTs is an interesting different avenue for work, potentially requiring further refinements and specialized techniques. For example, [2-6] are some examples of how complex and nuanced the space and applications can be.
> > > >
> > > > Finally, we note that [ICLR](https://iclr.cc/Conferences/2025#:~:text=Timezone%3A-,About%20Us,-The%20International%20Conference) has different areas for such papers, such as implementation issues and applications.
> > > >
> > > > > 3) See question 1). How can be sure that you improvement over the Naive-DP will not diminish with BatchNorm in the architecture?
> > > >
> > > >
> > > > To clarify, incorporating BatchNorm into either Naive-DP or Logit-DP would require a significant redesign to ensure privacy, as standard BatchNorm operations violate DP guarantees.  Developing novel DP-compatible BatchNorm layers is an interesting research direction but falls outside the scope of this work.
> > > > Does the reviewer have something different in mind?
> > > >
> > > >
> > > > >4)It is not uncommon to observe privacy leakage after a certain number of epochs.
> > > >
> > > > Could you please elaborate on what you mean by "privacy leakage"? Our epsilon value is calculated based on the total number of epochs, ensuring a privacy guarantee for the entire training process, i.e., all intermediate weights obtain the given level of differential privacy.
> > > >
> > > > > Training until convergence is a meaningful scenario, as it strengthens the validity of your findings.
> > > >
> > > > We have included results for a full training run to address this point (Figure 6).

---

> > ### Comment · Reviewer_2dkT · 2024-11-21
> >
> > Thank you for addressing my review and providing clarifications. In the following response I will try to reconsider your correction and look forward to further comments regarding the remaining concerns.
> >
> > **1)**
> > > we deliberately excluded BatchNorm layers from our models due to their inherent incompatibility with privacy preservation
> >
> > BatchNorm itself is a layer that harms privacy-preserving training. From our experiments it is unclear whether you achieve a private training / fin-tuning because of the Logit-DP or the absence of batch normalization. To my opinion, at least one simple experiment on this matter should be adjusted.
> >
> > **2)**
> > > LayerNorm's sample-wise operation is inherently more compatible with DP techniques
> >
> > Generally, I agree with this statement. While reading your work, I noted that there is quite a wide class of models you could consider, at least for the image classification tasks, such as ViTs and alike architectures. It seems more natural to focus on these transformer-based models rather than using "truncated" versions of ResNets (without BatchNorm), especially if you have decided to exclude batch normalization altogether.
> >
> > **3)**
> > > More specifically, we aim to showcase its potential for performance improvement within a constrained experimental setup.
> >
> > See question **1)**. How can be sure that you improvement over the Naive-DP will not diminish with BatchNorm in the architecture?
> >
> > **4)**
> > > These results, even with a limited number of training epochs, effectively illustrate the performance boost achieved by our method.
> > > This choice was deliberately made to prioritize efficiency in computational resources, especially given that this work focuses on the validation of our theoretical contributions.
> >
> > It is not uncommon to observe privacy leakage after a certain number of epochs. Training until convergence is a meaningful scenario, as it strengthens the validity of your findings.
> >
> > **5)**
> > > As mentioned above, our goal was to demonstrate the advantages of our proposed method, namely that we introduce a method that is able to overcome the loss stagnation of current contrastive loss methods, rather than training until state of the art results.
> >
> > No need to train to the state-of-the-art. Just to ensure that the Relative Loss Value for Logit-DP will be the best during the whole run. See **4)**

---

> ### Author Response · Authors · 2024-11-22
>
> > 5) No need to train to the state-of-the-art. Just to ensure that the Relative Loss Value for Logit-DP will be the best during the whole run. See 4)
>
> We have included these results (Figure 6) and demonstrate that Logit-DP consistently outperforms Naive-DP in terms of relative loss throughout the entire training process.
>
> > C2 is still unaddressed.
>
> We apologize for this oversight. We have corrected the title to accurately reflect "CIFAR100."
>
> > partially addressed W2 (regarding the bound on the plot at 180K examples mark) and have added a part of my correction to the text.
>
> We believe we have now fully addressed W2.
>
> >  The absence of at least one comparison with BatchNorm layers, one full training for 100+ epochs on CIFAR10
>
> These have now been included in the revised manuscript, see Table 3 and Figure 6 in the appendix.
>
> **Summary**
>
> We believe this revised manuscript comprehensively addresses the reviewer's concerns.  We have clarified the inherent incompatibility of standard BatchNorm with differential privacy, provided further experimental results as requested, and corrected the minor errors noted. We are confident that our work makes a valuable contribution to the privacy field and the unsupervised learning field by making progress on differentially private optimization of non-decomposable objective functions.
>
>
> **References**
>
> [1] Davody, A., Adelani, D. I., Kleinbauer, T., & Klakow, D. (2020). On the effect of normalization layers on differentially private training of deep neural networks. arXiv preprint arXiv:2006.10919.
>
> [2] Ponomareva, N., Hazimeh, H., Kurakin, A., Xu, Z., Denison, C., McMahan, H. B., ... & Thakurta, A. G. (2023). How to dp-fy ml: A practical guide to machine learning with differential privacy. Journal of Artificial Intelligence Research, 77, 1113-1201.
>
> [3] Dosovitskiy, A. (2020). An image is worth 16x16 words: Transformers for image recognition at scale. arXiv preprint arXiv:2010.11929.
>
> [4] Kong, W., & Munoz Medina, A. (2024). A unified fast gradient clipping framework for DP-SGD. Advances in Neural Information Processing Systems, 36.
>
> [5] Bu, Z., Mao, J., & Xu, S. (2022). Scalable and efficient training of large convolutional neural networks with differential privacy. Advances in Neural Information Processing Systems, 35, 38305-38318.
>
> [6] Lee, J., & Kifer, D. (2021). Scaling up differentially private deep learning with fast per-example gradient clipping. Proceedings on Privacy Enhancing Technologies.

---

> > ### Comment · Reviewer_2dkT · 2024-11-23
> >
> > Thank you for the insightful comments!
> >
> > > Finally, we note that ICLR has different areas for such papers, such as implementation issues and applications.
> >
> > Dear authors, as I see, the primary area for your work is: alignment, fairness, safety, privacy, and societal considerations. I believe that, at least for "alignment", "fairness", "safety", you should include a brief discussion of architectural ablations. And explain clearly in the text why you do not consider different architectural modifications such a LayerNorm after you decided not to use the BatchNorm.
> >
> > > Could you please elaborate on what you mean by "privacy leakage"?
> >
> > I meant the explosion of the loss given your privacy budget.
> >
> > > We have included results for a full training run to address this point (Figure 6).
> >
> > Thank you for your efforts. However, I notice that Figure 6 is incomplete: you do not show the Relative Loss Value for Logit-DP across all Number of Examples.
> >
> > > We have included these results (Figure 6) and demonstrate that Logit-DP consistently outperforms Naive-DP in terms of relative loss throughout the entire training process.
> >
> > Thank you. I believe that Logit-DP may be better than Naive-DP, but I think you should extend also Logit-DP experiment in this Figure.
> >
> >
> > > These have now been included in the revised manuscript, see Table 3 and Figure 6 in the appendix.
> >
> > Could you explain why the accuracy for the Non-private (-BN) outperforms the Non-private (+BN) model?
> >
> > **Summary:** I appreciate the authors' engagement during the discussion phase. However, my concern **W2** has not been fully addressed, specifically, Figure 6 still needs to be shown in its complete form.

---

> > > ### Author Response · Authors · 2024-11-28
> > >
> > > Thanks for the response and for carefully revising our revisions!
> > >
> > > First, we want to clarify that we uploaded the code to replicate our experiments as supplementary material addressing **W4**. We now address the more recent comments above.
> > >
> > > > And explain clearly in the text why you do not consider different architectural modifications such a LayerNorm after you decided not to use the BatchNorm.
> > >
> > > Thank you for the suggestion. While we did exclude BatchNorm to satisfy privacy requirements from DP, there was no specific reason why we intentionally excluded LayerNorm. LayerNorm could be a valuable addition for future investigation, but our primary goal in this work is to evaluate our DP approach within well-established architectures using common building blocks (dense and convolutional layers). This allows us to demonstrate the effectiveness of our method without confounding factors from novel architectural choices. A full exploration of architectural modifications, including LayerNorm, is an interesting direction for future research.
> > >
> > > > Could you please elaborate on what you mean by "privacy leakage"?
> > > > I meant the explosion of the loss given your privacy budget.
> > >
> > > Our experiments carefully manage the privacy budget to quantify this privacy leakage (which is fixed to epsilon=5.0), preventing the potential for a privacy loss explosion that can occur in non-DP models’ training.
> > >
> > >
> > > >Thank you for your efforts. However, I notice that Figure 6 is incomplete: you do not show the Relative Loss Value for Logit-DP across all Number of Examples…
> > > > Thank you. I believe that Logit-DP may be better than Naive-DP, but I think you should extend also Logit-DP experiment in this Figure…
> > >
> > > Extending the Logit-DP experiment to a larger number of examples is computationally expensive as noted in our manuscript. We believe the current plot is complete and validates our conclusions by clearly illustrating the rapid convergence of Logit-DP compared to Naive-DP, even within a limited number of examples.
> > >
> > > > Could you explain why the accuracy for the Non-private (-BN) outperforms the Non-private (+BN) model?
> > >
> > > Notice that while accuracy is better, all other metrics are slightly worse, and it is not generally true that the model without BatchNorm (-BN) outperforms the one with BatchNorm (+BN). It is true that results are very close.
> > > This can be due to the fact that BatchNorm doesn't always guarantee improved performance. While it can aid optimization stability, particularly in settings where doing careful hyperparameter tuning is hard, it's not universally beneficial across all datasets and architectures.
> > >
> > > In our case we did run careful hyperparameter tuning for non-private methods to be transparent about the results of private methods. Our findings, where the Non-private (-BN) model slightly outperforms the Non-private (+BN) model in terms of accuracy, are thus consistent with the above observation from the literature.
> > >
> > >
> > > > Summary: I appreciate the authors' engagement during the discussion phase. However, my concern W2 has not been fully addressed, specifically, Figure 6 still needs to be shown in its complete form.
> > >
> > > We thank the reviewer for their detailed feedback. We have carefully considered all of the reviewer's comments and suggestions, included our code,  and have made revisions to address each point, as detailed above. We believe these changes have strengthened the paper and clarified our contributions.

---

> > > > ### Comment · Reviewer_2dkT · 2024-12-02
> > > >
> > > > Dear authors
> > > >
> > > > Thanks for the active response over the last week.
> > > > I read your revision and noticed that you still have not revisited Figure 6.
> > > >
> > > > Therefore, I will keep my score.

---

> > > > > ### Author Response · Authors · 2024-12-03
> > > > >
> > > > > Dear reviewer,
> > > > >
> > > > > Thank you for your feedback and active discussion throughout the rebuttal period. We believe Figure 6 adequately illustrates the rapid convergence of Logit-DP compared to Naive-DP, even within a limited number of examples, supporting our conclusions.
> > > > >
> > > > > As noted in our manuscript, and previous comments, extending the Logit-DP experiment to a larger number of examples is computationally expensive. We kindly remind you that ICLR guidelines discourage requests for significant additional experiments.

---

### Official Review · Reviewer_EWmR · 2024-11-07

**Soundness:** 2
**Presentation:** 3
**Contribution:** 2
**Rating:** 6
**Confidence:** 4

**Summary:**

The paper presents a new variant of Differentially Private Stochastic Gradient Descent (DP-SGD) designed for similarity-based loss functions, such as contrastive loss, which are common in unsupervised pre-training. The core claimed contribution is a modified DP-SGD method that achieves sensitivity of $O(1)$ for the summed gradient with respect to batch size. The paper also provides experimental validation of this new method on CIFAR-10 and CIFAR-100, showing performance close to non-private models and generally better than the naive DP-SGD approach.

**Strengths:**

- The problem is very well-motivated. Differentially private pre-training is an important and interesting direction for learning foundation models privately. Most existing work [PBV2022, YSM+2024] focus on training decomposable loss with DP-SGD. Investigating new efficient and effective techniques for training DP models with non-decomposable loss is very important.


*References*

[PBV2022] Training text-to-text transformers with privacy guarantees. Ponomareva, N., Bastings, J., and Vassilvitskii, S.  In Findings of the Association for Computational Linguistics: ACL 2022, pp. 2182–2193, 2022.

[YSM+2024] ViP: A Differentially Private Foundation Model for Computer Vision. Yaodong Yu, Maziar Sanjabi, Yi Ma, Kamalika Chaudhuri, Chuan Guo. Proceedings of the 41st International Conference on Machine Learning, PMLR 235:57639-57658, 2024.

**Weaknesses:**

The main question (and potential weakness) I have is:
>Is the proposed method indeed independent of the batch size $n$? As proved in Theorem 4.2, the $L_2$ sensitivity is upper bounded by $(G_1 + G_2 + (n-1)L) B$, where $n$ is the batch size. Then as described in Lemma 4.5, for contrastive loss, $(G_1 + G_2 + (n-1)L) B$ is upper bounded by $B_{\text{contrastive}}=2(1+\frac{(n-2)e^2}{e^2+(n-1)})$. Therefore, when $n$ is large, isn't $B_{\text{contrastive}} = O(n)$? And this is not independent of the batch size $n$.

I may have some misunderstandings here. I would like have the authors to clarify this during the discussion. If the independent argument is correct, I would raise my score.

(minor weakness) The proposed method is computationally more expensive than standard DP-SGD, I suggest the authors to provide some results on [training loss vs training time].

**Questions:**

A few missing references on unsupervised DP-SGD pre-training:

[PBV2022] Training text-to-text transformers with privacy guarantees. Ponomareva, N., Bastings, J., and Vassilvitskii, S.  In Findings of the Association for Computational Linguistics: ACL 2022, pp. 2182–2193, 2022.

- [YSM+2024] ViP: A Differentially Private Foundation Model for Computer Vision. Yaodong Yu, Maziar Sanjabi, Yi Ma, Kamalika Chaudhuri, Chuan Guo Proceedings of the 41st International Conference on Machine Learning, PMLR 235:57639-57658, 2024.

---

> ### Author Response · Authors · 2024-11-20
> **Response to reviewer EWmR**
>
> Thank you for the review!
>
>   1. You're correct that a naive approach would result in $B_{contrastive}$ being $O(n)$. However, in our proposed method, as the batch size $n$ grows, $B_{contrastive}$ converges to the constant $e^2$. As shown in (6), this is because $n$ is both in the numerator and denominator and the dominant term in $B_{contrastive}$ becomes independent of $n$ for large values. This contrasts with the naive scheme, where $B_{contrastive}$ does indeed grow linearly with $n$.
>
>
>   2. We now provide these graphs in Figure 5 in the appendix. While Logit-DP is computationally more expensive than Naive DP-SGD per-iteration, it is more efficient in terms of loss decay, as shown in Figure 5.
>
> *Questions*
>
> - We've incorporated your suggested missing references in the revised version, in Section 3, Related Work.
>
>   In brief, Ponomareva et al. focus on efficient private pre-training for T5, which doesn't need usage of a non-decomposable loss. Yu et al. uses a model typically trained with a contrastive loss in non-private settings, but their approach substitutes it with a per-example loss. This leads to a loss of information from unlabeled data points and inter-sample correlations.

---

> > ### Comment · Reviewer_EWmR · 2024-11-28
> > **Response to rebuttal**
> >
> > Thank the authors for the rebuttal.
> >
> > For 1. I apologize that I made a trivial mistake, and $B_{\text{contrastive}} \rightarrow 2+e^2$.
> >
> > For 2. Thank the authors adding the new experiments regarding computational time, which could be helpful for understanding the efficiency of the proposed method.
> >
> > My questions are well addressed, and I have increased my score.

---

### Author Response · Authors · 2024-11-20
**Overall Remarks**

We thank all the reviewers for the helpful comments! We have updated the manuscript by addressing them, with changes highlighted in purple. We do not provide pointers to our code for anonymity reasons, but we plan to add a link with a camera-ready version. In addition and in response to reviewer 2dkT's comment about our code, we are working on an anonymized version that we plan to upload within the next few days.

We provide individual reviewer responses below.

---

> ### Author Response · Authors · 2024-11-28
> **Code available**
>
> We have incorporated the suggested changes and clarifications into the revised manuscript. Additionally, the code associated with this work is now available in the supplementary material.

---

### Meta-Review · Area_Chair_fCgh · 2024-12-19

**Metareview:**

The paper introduces a new variant of Differentially Private Stochastic Gradient Descent (DP-SGD) tailored for unsupervised pretraining losses such as contrastive loss. The key contribution is the observation that log-sum-exponential loss used for contrastive learning has gradient sensitivity that doesn't scale with batchszie if clipped in the right way. However, the proposed method is also considerably more computationally expensive than non-private method and requires non-trivial changes to the backprop mechanism. Most reviewers noted that the claim is correct, but it has limited technical novelty. Reviewers also noted issues with lack of clarity, rigor and comprehensiveness of the experiments.

**Additional Comments On Reviewer Discussion:**

Authors managed to improve clarity and add more experimental results during the discussion phase. However, most reviewers did not change their evaluation after this exchange.

---

### Decision · Program_Chairs · 2025-01-22

Accept (Poster)